# Time-Dependent Pseudo-Hermitian Hamiltonians and a Hidden Geometric Aspect of Quantum Mechanics

**DOI:** 10.3390/e22040471

**Published:** 2020-04-20

**Authors:** Ali Mostafazadeh

**Affiliations:** Departments of Mathematics and Physics, Koç University, Sarıyer, 34450 Istanbul, Turkey; amostafazadeh@ku.edu.tr

**Keywords:** pseudo-Hermitian operator, time-dependent Hilbert space, energy observable, Heisenberg picture

## Abstract

A non-Hermitian operator *H* defined in a Hilbert space with inner product 〈·|·〉 may serve as the Hamiltonian for a unitary quantum system if it is η-pseudo-Hermitian for a metric operator (positive-definite automorphism) η. The latter defines the inner product 〈·|η·〉 of the physical Hilbert space $H_{\eta }$ of the system. For situations where some of the eigenstates of *H* depend on time, η becomes time-dependent. Therefore, the system has a non-stationary Hilbert space. Such quantum systems, which are also encountered in the study of quantum mechanics in cosmological backgrounds, suffer from a conflict between the unitarity of time evolution and the unobservability of the Hamiltonian. Their proper treatment requires a geometric framework which clarifies the notion of the energy observable and leads to a geometric extension of quantum mechanics (GEQM). We provide a general introduction to the subject, review some of the recent developments, offer a straightforward description of the Heisenberg-picture formulation of the dynamics for quantum systems having a time-dependent Hilbert space, and outline the Heisenberg-picture formulation of dynamics in GEQM.

## 1. Introduction

The fact that a non-Hermitian operator can have a real spectrum is by no means unusual or surprising. For example, consider the operator $H:C^{2}\to C^{2}$ that is represented in the standard basis of $C^{2}$ by the matrix
$$H:=\varepsilon \left[\begin{array}{cc} 0 & 1\\ 4 & 0 \end{array}\right],$$
where ε is a positive real parameter. It is easy to check that H and consequently *H* have a pair of real eigenvalues namely ±2ε. In particular, they are diagonalizable and have a real spectrum, but does this mean that we can identify *H* with an observable or the Hamiltonian of a quantum system? The answer to this question cannot be given unless we specify the inner product we wish to use for computing the expectation values of the observables of the system. If we adopt the standard Euclidean inner product 〈·|·〉, the answer is No. To see this, we recall that by definition, $\langle \xi |\zeta \rangle :=\xi_{1}^{*}\zeta_{1}+\xi_{2}^{*}\zeta_{2}$, where $\xi :=(\xi_{1},\xi_{2})$ and $\zeta :=(\zeta_{1},\zeta_{2})$ are arbitrary elements of $C^{2}$, use H to label the Hilbert space obtained by endowing $C^{2}$ with the Euclidean inner product, and calculate the expectation value of *H* in the state determined by the state vector $\chi :=\frac{1}{\sqrt{2}}\left(1,-i\right)$. This gives
(1)$$\frac{\langle \chi |H\chi \rangle }{\langle \chi |\chi \rangle }=\frac{3i\varepsilon }{2}.$$
Because this quantity is purely imaginary, we cannot interpret it as the average value of measurement outcomes ±2ε which are real. This disqualifies *H* to represent an observable of a quantum system with Hilbert space H, if we are to respect the measurement (projection) axiom of quantum mechanics (QM) [1,2].

The fact that *H* can have a complex expectation value is a manifestation of a basic result of linear algebra [3,4] which says: “*A linear operator is Hermitian if and only if all its expectation values are real.*” Because the reality of the expectation values is an indispensable ingredient of the measurement axiom, the claim that observables of a quantum system need not be Hermitian is false (Throughout this article we distinguish between operators and their matrix representations, for the latter depends on the choice of a basis. In particular, following von Neumann [2], we use the term “Hermitian operator” to mean “self-adjoint operator”, i.e., *H* satisfies 〈·,H·〉=〈H·,·〉, where 〈·,·〉 denotes the inner product of the Hilbert space. For a more precise definition see [5]).

Another better known problem arises, if we try to identify *H* with the Hamiltonian of a quantum system with Hilbert space H, i.e., demand that it generates the dynamics of the system via the Schrödinger equation,
(2)$$i\partial_{t}\psi \left(t\right)=H\psi \left(t\right).$$
According to this equation,
(3)∂t〈ψ(t)|ψ(t)〉=〈∂tψ(t)|ψ(t)〉+〈ψ(t)|∂tψ(t)〉=2Re〈ψ(t)|∂tψ(t)〉=2Im〈ψ(t)|Hψ(t)〉,
where “Re” and “Im” denote the real and imaginary part of their argument. Because *H* has non-real expectation values the right-hand side of this equation can be nonzero. For example, letting ψ(0):=χ and using (Equation 1) and (Equation 4), we find $\partial_{t}\langle \psi \left(t\right)|\psi \left(t\right)\rangle {|}_{t=0}=3\varepsilon \neq 0$. This shows that the norm of the evolving state does change in time. Hence, *H* does not generate a unitary time evolution.

The apparent conflicts with the measurement and unitarity axioms were responsible for the unpopularity of non-Hermitian operators among physicists interested in basic aspects of QM. For many decades their application was confined to effective theories which did not respect all of the Dirac-von Neumann axioms of QM. This situation drastically changed in the early 2000’s after it was realized that a certain class of non-Hermitian operators can actually be made Hermitian upon a redefinition of the inner product of the Hilbert space [6,7,8,9,10,11,12]. The operator *H* we considered above is a particular example. Let $\eta :C^{2}\to C^{2}$ and ${\langle \zeta ,\xi \rangle }_{\eta }:C^{2}\times C^{2}\to C$ be defined by
(4)$$\begin{array}{c} \eta \;\xi =\eta \left(\xi_{1},\xi_{2}\right):=\left(\xi_{1},\frac{\xi_{2}}{4}\right), \end{array}$$
(5)$$\begin{array}{c} {\langle \zeta ,\xi \rangle }_{\eta }:=\langle \zeta |\eta \;\xi \rangle :=\zeta_{1}^{*}\xi_{1}+\frac{\zeta_{2}^{*}\xi_{2}}{4}. \end{array}$$
Then, ${\langle \cdot ,\cdot \rangle }_{\eta }$ defines a genuine (positive-definite) inner product [5] in $C^{2}$, and for every nonzero element $\xi =(\xi_{1},\xi_{2})$ of $C^{2}$, we have
〈ξ,Hξ〉η〈ξ,ξ〉η=8εRe(ξ1*ξ2)4|ξ1|2+|ξ2|2.
This calculation shows that the expectation values of *H* computed using the inner product ${\langle \cdot ,\cdot \rangle }_{\eta }$ are real. Therefore, if we view *H* as a linear operator acting in the Hilbert space $H_{\eta }$ defined by endowing $C^{2}$ with the inner product ${\langle \cdot ,\cdot \rangle }_{\eta }$, then it becomes Hermitian, i.e.,
$${\langle \zeta ,H\xi \rangle }_{\eta }={\langle H\zeta ,\xi \rangle }_{\eta }.$$
This in turn implies the unitarity of the dynamics generated by the Schrödinger Equation (Equation 2) in the Hilbert space $H_{\eta }$, i.e., for each pair, ϕ(t) and ψ(t), of solutions of this equation,
$$\partial_{t}\;{\langle \phi \left(t\right),\psi \left(t\right)\rangle }_{\eta }=0.$$

The operator η given by (Equation 4) is an example of a metric operator acting in the Hilbert space H. We use the term “metric operator” to mean a positive-definite authomorphism (a positive-definite one-to-one linear operator mapping all of H onto H). This property ensures ${\langle \cdot ,\cdot \rangle }_{\eta }$ to be a genuine positive-definite inner product. The requirement that *H* is a Hermitian operator acting in $H_{\eta }$ is equivalent to demanding that it acts in H as an η-pseudo-Hermitian operator, i.e.,
(6)$$H^{†}=\eta \;H\;\eta^{-1},$$
where $H^{†}$ is the adjoint of *H* viewed as an operator acting in H. The latter is defined by the condition: $\langle \zeta |H^{†}\xi \rangle =\langle H\zeta |\xi \rangle$. We can also view *H* as an operator acting in $H_{\eta }$ and introduce its adjoint $H^{♯}$ through the requirement: ${\langle \zeta ,H^{♯}\xi \rangle }_{\eta }={\langle H\zeta ,\xi \rangle }_{\eta }$. It is not difficult to see that this is equivalent to
$$H^{♯}:=\eta^{-1}H^{†}\eta .$$
In light of this relation, we can identify (Equation 6) with $H^{♯}=H$, [6]. Therefore, η-pseudo-Hermitian operators acting in H coincide with Hermitian operators acting in $H_{\eta }$. These constitute the observables of the quantum system determined by the Hilbert space-Hamiltonian operator pair $\left(H_{\eta },H\right)$, [12].

The notion of a pseudo-Hermitian operator as defined by (Equation 6) extends to situations where η is a pseudo-metric operator, i.e., it is a Hermitian automorphism that needs not be positive-definite. In this more general setting and under the assumption that *H* acts in a given Hilbert space H, has a discrete spectrum, and is diagonalizable (i.e., has a complete and bounded biorthonormal system [5] formed out of its eigenvectors and those of its adjoint), one can prove that the following statements are equivalent [8].
(1)*H* is η-pseudo-Hermitian for a pseudo-metric operator η, i.e., it satisfies (Equation 6).(2)The eigenvalues of *H* are either real or come in complex-conjugate pairs.(3)There is an antilinear operator X that squares to identity and commutes with *H*.
For situations where *H* is expected to play the role the Hamiltonian of a quantum system the latter statement means that X generates an antilinear symmetry of the system [13]. This in turn clarifies the spectral consequences of PT-symmetry [14,15,16,17].

With the stronger requirement that η be positive-definite one can establish the reality of the spectrum of *H*, its quasi-Hermiticity (existence of a positive-definite automorphsim ρ such that $h:=\rho^{-1}H\;\rho$ is Hermitian [18]), and the exactness of the antilinear symmetry X. More precisely the following statements are equivalent [8,11].

(1${}^{\prime }$)*H* is η-pseudo-Hermitian for a metric operator η.(2${}^{\prime }$)*H* acts as a Hermitian operator in $H_{\eta }$.(3${}^{\prime }$)The eigenvalues of *H* are real.(4${}^{\prime }$)The operator $h:=\rho^{-1}H\;\rho$ with $\rho :=\sqrt{\eta }$ acts as a Hermitian operator in H, where $\sqrt{\eta }$ stands for the positive square root of η. In particular as an operator acting in H, *H* is quasi-Hermitian.(5${}^{\prime }$)There is an antilinear operator X that squares to identity, and there is a complete set of common eigenvectors of *H* and X.

Suppose that the statement 1${}^{\prime }$ holds, so that $H:H_{\eta }\to H_{\eta }$ is Hermitian. Then we can identify $H_{\eta }$ and *H* with the Hilbert space and Hamiltonian of a quantum system S. Being a Hermitian operator acting in $H_{\eta }$, *H* determines an observable of S. Furthermore, because Hermitian operators have real expectation values, a calculation similar to the one leading to (Equation 4) implies that *H* generates unitary evolutions. Hence S is a unitary quantum system. An alternative way of arriving at this conclusion is to note that the Hilbert space-Hamiltonian operator pair (H,h) also describes the same quantum system S. To see this first we recall the definition of a unitary operator.

In standard texts on quantum mechanics, a unitary operator U is defined as a linear mapping that maps a given Hilbert space onto the same Hilbert space and preserves the inner product of vectors. There is a standard generalization of this notion to the case that the operator U maps a Hilbert space $H_{1}$ with inner product ${\langle \cdot ,\cdot \rangle }_{1}$ onto another Hilbert space $H_{2}$ with inner product ${\langle \cdot ,\cdot \rangle }_{2}$. If the domain of U is $H_{1}$, and for all $\psi_{1},\phi_{1}\in H_{1}$ we have ${\langle \psi_{1},\phi_{1}\rangle }_{1}={\langle U\psi_{1},U\psi_{2}\rangle }_{2}$, we say that U is a unitary operator. According to this definition, ρ defines a unitary operator mapping $H_{\eta }$ to H, [11], because
$$\langle \rho \;\zeta |\rho \;\xi \rangle =\langle \zeta |\rho^{2}\xi \rangle =\langle \zeta |\eta \;\xi \rangle ={\langle \zeta ,\xi \rangle }_{\eta }.$$
This in turn implies that if $\psi \in H_{\eta }$ and $O:H_{\eta }\to H_{\eta }$ respectively describe a state and an observable of S, Ψ:=ρψ∈H and $o:=\rho \;O\rho^{-1}:H\to H$ describe the same state and observable of S. This is simply because both choices lead to the same expectation values;
$$\frac{{\langle \psi ,O\psi \rangle }_{\eta }}{{\langle \psi ,\psi \rangle }_{\eta }}=\frac{\langle \psi |\eta O\psi \rangle }{\langle \psi |\eta \psi \rangle }=\frac{\langle \psi |\rho^{2}O\psi \rangle }{\langle \psi |\rho^{2}\psi \rangle }=\frac{\langle \rho \psi |o\rho \psi \rangle }{\langle \rho \psi |\rho \psi \rangle }=\frac{\langle \Psi |o\;\Psi \rangle }{\langle \Psi |\Psi \rangle }.$$
This shows that $\left(H_{\eta },H\right)$ and (H,h) provide different mathematical representations of the same quantum system [12]. In particular, we can use either of them to determine the physical properties of this system.

The initial work on pseudo-Hermitian operators [6,7,8] was motivated by the need for a careful evaluation of the prospects of PT-symmetric QM [15] and the possible relevance of these operators to certain constructions arising in the two-component formulation of the mini-superspace Wheeler-DeWitt equation [19].

The results reported in Refs. [11,12,20] showed that indeed certain PT-symmetric Hamiltonian operators were capable of defining unitary quantum systems, but these systems also admitted a description in terms of Hermitian Hamiltonian operators. Therefore, the use of PT-symmetric (and more generally pseudo-Hermitian) Hamiltonians do not actually yield a generalization of QM. It rather gives rise to previously unexplored equivalent representations of quantum mechanics [5].

An important by-product of the study of pseudo-Hermitian operators was the introduction of new technologies for the construction of inner products [6,7,21]. For certain physically interesting quantum cosmological models, these could be employed for the purpose of endowing the solution space of the Wheeler-DeWitt equation with the structure of a genuine Hilbert space [22,23]. This meant solving the infamous Hilbert-space problem [24] for these models. The same approach allowed for a complete and consistent formulation of QM of a first-quantized free Klein-Gordon field [25,26,27], a Proca field [28], and more recently a free photon [29,30].

Quantum cosmological applications of pseudo-Hermitian operators require dealing with time-dependent metric operators [22,23]. For a quantum system represented by the Hilbert space-Hamiltonian operator pair $\left(H_{\eta },H\right)$, the proof of the unitarity of time-evolution encounters a major difficulty whenever η depends on time. More precisely, the requirement of unitarity of dynamics conflicts with the η-pseudo-Hermiticity and hence observability of the Hamiltonian. Since its announcement [31] in 2007, there have appeared different proposals for resolving this conflict in the literature [32,33,34,35,36,37]. A careful assessment of the geometric aspects of this problem has recently led to a comprehensive resolution that not only clarifies the role of the energy operator for quantum systems having a dynamical Hilbert space, but also paves the way towards a geometric extension of quantum mechanics (GEQM) [38]. In the present article, we provide a brief review of these developments, discuss their conceptual implications, and outline a Heisenberg-picture formulation of the dynamics for systems with a time-dependent state space and systems considered in the framework of GEQM.

## 2. Time-Dependent Pseudo-Hermiticity

Consider a quantum system S represented by the Hilbert space-Hamiltonian operator pair $\left(H_{\eta },H\right)$, where η is a time-dependent metric operator, and let ψ and ϕ be arbitrary solutions of the Schrödinger Equation (Equation 2). Then,
(7)$$\begin{array}{ccc} \partial_{t}{\langle \phi |\psi \rangle }_{\eta } & = & \partial_{t}\langle \phi |\eta \;\psi \rangle \\ & = & \langle \dot{\phi }|\eta \;\psi \rangle +\langle \phi |\eta \;\dot{\psi }\rangle +\langle \phi |\dot{\eta }\;\psi \rangle \\ & = & \langle -iH\phi |\eta \;\psi \rangle +\langle \phi |-i\eta \;H\psi \rangle +\langle \phi |\dot{\eta }\;\psi \rangle \\ & = & i\langle \phi |\left(H^{†}\eta -\eta \;H-i\dot{\eta }\right)\psi \rangle \\ & = & i{\langle \phi ,\left(H^{♯}-H-i\eta^{-1}\dot{\eta }\right)\psi \rangle }_{\eta }, \end{array}$$
where an overdot labels a time derivative. In order for *H* to generate a unitary evolution, the right-hand side of (Equation 8) must vanish for every choice of the solutions ϕ and ψ. This happens if and only if
(8)$$H^{♯}=H+i\eta^{-1}\dot{\eta }.$$
Because η is time-dependent and $\eta^{-1}$ is invertible, this equation implies, $H^{♯}\neq H$, i.e., *H* is not a Hermitian operator acting in $H_{\eta }$. Therefore, if *H* generates a unitary dynamics, it does not correspond to an observable of the quantum system S! This is the content of the conflict between the unitarity of the time evolution generated by the Schrödinger Equation (Equation 2) in $H_{\eta }$ and the observability of the Hamiltonian *H*, [31].

The initial work on the construction of the most general metric operator η for a diagonalizable Hamiltonian *H* with a real and discrete spectrum [6,7] revealed the following spectral expansion of η.
(9)$$\eta =\sum_{n}|\phi_{n}\rangle \langle \phi_{n}|,$$
where $\phi_{n}$ are eigenvectors of $H^{†}$ that constitute a (Riesz) basis of the Hilbert space [5], and for every ζ∈H, the symbol |ζ〉〈ζ| stands for the linear operator that maps state vectors ξ to 〈ζ|ξ〉ζ. A simple consequence of (Equation 9) is that unless *H* and therefore $H^{†}$ have a complete set of time-independent eigenvectors, every metric operator η that renders *H* pseudo-Hermitian is necessarily time-dependent. This underlines the significance of addressing the conflict between the observability of generic time-dependent Hamiltonians and the unitarity of the dynamics they generate.

There are essentially three different ways of dealing with this conflict:(i)Modifying the Schrödinger equation to avoid this conflict.(ii)Upholding unitarity at the expense of unobservability of the Hamiltonian.(iii)Abandoning the requirement of unitarity in favor of the observability of the Hamiltonian.
To the best of our knowledge option iii was never considered as viable, while there appeared a number of publications [32,33,34,35,36,37] advocating options i or ii. The developments reported in these publications rest on the following premises:(a)There is a representation of S defined by the Hilbert space H and a generally time-dependent Hermitian Hamiltonian operator *h* acting in H. This operator generates the dynamics of the state vectors in H via the standard Schrödinger equation,
(10)$$i\partial_{t}\Psi \left(t\right)=h\left(t\right)\Psi \left(t\right),$$
and identifies an observable of the system which is customarily called the *energy observable*.(b)Given a possibly time-dependent metric operator η, we can represent S using the Hilbert space $H_{\eta }$ and an operator *H* that generates time evolutions in $H_{\eta }$, such that the unitary transformation $\rho^{-1}:H\to H_{\eta }$ maps the solutions of the Schrödinger Equation (Equation 10) defined by *h* to those of the Schrödinger Equation (Equation 2) defined by *H*. It is easy to show that this condition is equivalent to the requirement:
(11)$$H=\rho^{-1}h\rho -i\rho^{-1}\dot{\rho }.$$(c)In the representation $\left(H_{\eta },H\right)$, the observables of S, which are represented by Hermitian operators *O* acting in $H_{\eta }$, are obtained from their representatives *o* in the representation (H,h) via $O=\rho^{-1}o\rho$. In particular, in the representation $\left(H_{\eta },H\right)$, the energy observable is represented by
(12)$$H_{E}:=\rho^{-1}h\rho .$$

If we insist that the Hamiltonian and the energy observable must coincide in both of the representations, (H,h) and $\left(H_{\eta },H\right)$, we have no choice but to agree that, in the representation $\left(H_{\eta },H\right)$, the dynamical evolution of the state vectors is determined by the modified Schrödinger equation [32,34],
(13)$$iD_{t}\psi =H_{E}\psi ,$$
where
(14)$$D_{t}:=\partial_{t}+\rho^{-1}\dot{\rho }.$$
This provides a resolution of the unitarity versus observability conflict via a modification of the Schrödinger equation. Note, however, that this approach stems from a particular choice of terminology. We could simply refrain from using the term “Hamiltonian” for the “energy operator”, but instead take the former to mean the “generator of time evolutions” determined by the usual Schrödinger Equation (Equation 2). We are then led to the inevitable conclusion that the Hamiltonian is not an observable unless ρ and consequently η are time-independent [36]. This is in line with the resolution ii of the above-mentioned conflict.

## 3. Dynamical Inner Products Realizing Unitarity

In specific applications in quantum cosmology [22,23], the generator of time evolutions is the only input of the problem, and the aim is to determine an appropriate Hilbert space in which the time evolution is realized via a one-parameter family of unitary operators. If one can identify a Hilbert space H in which the generator of time evolutions acts as a linear operator with a real and discrete spectrum and there is complete and bounded biorthonormal system [5] consisting of the eigenvectors of this operator and its adjoint, then there are metric operators η such that this operator is η-pseudo-Hermitian. However, for cases where all the metric operators η with this property are time-dependent, we cannot establish the unitarity of the time evolution by working in the Hilbert space $H_{\eta }$. Ref. [22] offers a solution for this problem that involves finding metric operators η that achieve the unitarity of the time evolutions, not the η-pseudo-Hermiticity of their generator.

Let H(t) label the generator of time evolutions, and $U(t,t_{0})$ be the corresponding evolution operator for the initial time $t_{0}$, so that $i\partial_{t}U\left(t,t_{0}\right)=H\left(t\right)U\left(t,t_{0}\right)$ and $U(t_{0},t_{0})=I$, where *I* is the identity operator acting in H. We can express the unitarity of dynamical evolutions in $H_{\eta (t)}$ in the form
$${\langle \phi \left(t\right),\psi \left(t\right)\rangle }_{\eta (t)}={\langle \phi \left(t_{0}\right),\psi \left(t_{0}\right)\rangle }_{\eta (t_{0})}.$$
This relation implies that for every choice of initial state vectors $\phi \left(t_{0}\right):=\phi_{0}$ and $\psi \left(t_{0}\right):=\psi_{0}$,
$$\begin{array}{ccc} \langle \phi_{0}|\eta \left(t_{0}\right)\psi_{0}\rangle & = & \langle \psi (t)|\eta (t)\psi (t)\rangle \\ & = & \langle U\left(t,t_{0}\right)\phi_{0}|\eta \left(t\right)U\left(t,t_{0}\right)\psi_{0}\rangle \\ & = & \langle \phi_{0}|U{\left(t,t_{0}\right)}^{†}\eta \left(t\right)U\left(t,t_{0}\right)\psi_{0}\rangle . \end{array}$$
This is true for every $\phi_{0},\psi_{0}\in H$ if and only if
(15)$$\eta \left(t\right)={U{\left(t,t_{0}\right)}^{†}}^{-1}\eta_{0}\;U{\left(t,t_{0}\right)}^{-1},$$
where $\eta_{0}:=\eta \left(t_{0}\right)$. Equation (Equation 15) determines the metric operator η(t) and consequently $H_{\eta (t)}$ up to the choice of $\eta_{0}$. A suitable choice, which is however not dictated by the details of the problem at hand, is to identify $\eta_{0}$ with a metric operator so that $H(t_{0})$ is $\eta_{0}$-pseudo-Hermitian [22,23]. This in turn implies that $H(t_{0})$ is an observable of the system represented by $\left(H_{\eta (t)},H\left(t\right)\right)$ at time $t_{0}$, but for $t\neq t_{0}$ the same does not generally apply to H(t). Notice however that there is a priori no reason to assume that $H(t_{0})$ is $\eta_{0}$-pseudo-Hermitian for some metric operator $\eta_{0}$. According to (Equation 8) whenever such a metric operator exists, the choice $\eta \left(t_{0}\right)=\eta_{0}$ is equivalent to $\dot{\eta }\left(t_{0}\right)=0$.

An important observation regarding (Equation 15) is that it provides the general solution of (Equation 8) when we view the latter as an equation for η. Using this equation, we can actually check that
(16)$$h\left(t\right):=\rho \left(t\right)H\left(t\right)\rho {\left(t\right)}^{-1}+i\dot{\rho }\left(t\right)\rho {\left(t\right)}^{-1}$$
is a Hermitian operator acting in H. Furthermore, because it satisfies (Equation 11), $\rho \left(t\right):=\sqrt{\eta (t)}$ maps the solutions of the Schrödinger Equation (Equation 2) for the Hamiltonian H(t) to those of the Schrödinger Equation (Equation 10) for the Hamiltonian h(t). By virtue of the fact that $\rho \left(t\right):H_{\eta (t)}\to H$ is a unitary operator, this shows that (H,h(t)) and $\left(H_{\eta (t)},H\left(t\right)\right)$ represent the same quantum system. A rather unexpected aspect of the latter representation is that not only H(t) fails to be η(t)-pseudo-Hermitian, but indeed it may happen not to be a pseudo-Hermitian operator at all, i.e., there may exist no metric operator $\tilde{\eta }\left(t\right)$ such that H(t) is $\tilde{\eta }\left(t\right)$-pseudo-Hermitian.

As a simple example, consider the situation where H is the Hilbert space of square-integrable functions and
(17)$$H\left(t\right):=H_{0}\left(t\right)+f\left(t\right)P,$$
where $H_{0}:=P^{2}/2m+m\omega^{2}X^{2}/2$ is the standard Hamiltonian for a simple harmonic oscillator with mass *m* and angular frequency ω, *X* and *P* are the standard position and momentum operators acting in H, f:R→C is a piecewise continous complex-valued function of time, and P is the parity operator defined by (Pψ)(x):=ψ(−x).

Because $H_{0}$ and P act in H as commuting Hermitian operators, the spectrum of H(t) consists of the eigenvalues of the form ω(n+1/2)±f(t), where *n* is a nonnegative integer. This shows that for the cases where f(t) is neither real nor imaginary, H(t) is not pseudo-Hermitian. Yet we can compute its evolution operator and use (Equation 15) to determine a metric operator that makes the time evolution generated by H(t) unitary. Setting $\eta_{0}=I$, so that $H_{\eta (t_{0})}=H$, we find
(18)U(t,t0)=U0(t,t0)e−iF(t)P,η(t)=e−2Im[F(t)]P,ρ(t)=e−Im[F(t)]P,
where $U_{0}\left(t,t_{0}\right):=\exp\left[-i\left(t-t_{0}\right)H_{0}\right]$ is the time-evolution operator for the simple harmonic oscillator, and $F\left(t\right):=\int_{t_{0}}^{t}f\left(t^{\prime }\right)dt^{\prime }$. Substituting (Equation 17) in (Equation 16) and using the last relation in (Equation 18), we have
h(t)=H0+Re[f(t)]P.
This shows that the quantum system represented by $\left(H_{\eta (t)},H\left(t\right)\right)$ also admits the representation (H,h(t)).

If we identify h(t) with the energy observable of the system in the representation (H,h(t)), then in view of (Equation 12) and (Equation 18) the operator $H_{E}\left(t\right)$ representing this observable in $\left(H_{\eta (t)},H\left(t\right)\right)$ coincides with h(t). This is not generally true for other observables. For example, in the representation $\left(H_{\eta (t)},H\left(t\right)\right)$, the position and momentum operators are given by [39]: xη:=ρ(t)−1Xρ(t)=e2Im[F(t)]PX,pη:=ρ(t)−1Pρ(t)=e2Im[F(t)]PP.

If we insist on using the term “Hamiltonian” for the energy operator $H_{E}$ and demand that this operator generates the dynamics via a first-order linear differential equation involving $H_{E}$, we are led to the modified Schrödinger Equation (Equation 13) with $H_{E}\left(t\right)=h\left(t\right)$ and
Dt:=∂t−Im[f(t)]P.

## 4. Heisenberg Picture of Dynamics

The description of the dynamics of a quantum system in the Heisenberg picture has many advantages. The study of the Heisenberg picture for a unitary quantum system defined by a time-independent pseudo-Hermitian Hamiltonian or a Hamiltonian acting in a time-dependent Hilbert space has been considered in Refs. [35,40]. In this section we provide our approach for addressing this problem.

Consider the representation (H,h(t)) of our generic quantum system S where observables are given by Hermitian operators o(t):H→H, and the dynamics of state vectors is generated by the Hermitian Hamiltonian operator h(t). In the Heisenberg picture, the state vectors are stationary while the operators corresponding to observables evolve in time according to
(19)$$o\left(t_{0}\right)⟶o^{\left(H\right)}\left(t\right):=u{\left(t,t_{0}\right)}^{-1}o\left(t\right)u\left(t,t_{0}\right).$$
Here $u(t,t_{0})$ is the time-evolution operator corresponding to the Hamiltonian h(t) and the initial time $t_{0}$, i.e., the operator satisfying
(20)$$\begin{array}{cccc} \; & i\partial_{t}u\left(t,t_{0}\right)=h\left(t\right)u\left(t,t_{0}\right),\; & \; & u(t_{0},t_{0})=I. \end{array}$$
If we differentiate both sides of (Equation 19) and use (Equation 20) to simplify the result, we obtain the Heisenberg equation of motion in the representation (H,h(t)): (21)$$i\partial_{t}o^{\left(H\right)}\left(t\right)=\left[o^{\left(H\right)}\left(t\right),h^{\left(H\right)}\left(t\right)\right]+iu{\left(t,t_{0}\right)}^{-1}\dot{o}\left(t\right)u\left(t,t_{0}\right),$$
where $h^{\left(H\right)}\left(t\right):=u{\left(t,t_{0}\right)}^{-1}h\left(t\right)u\left(t,t_{0}\right)$ is the Heisenberg-picture Hamiltonian.

Next, we examine the Heisenberg equation in the representation $\left(H_{\eta (t)},H\left(t\right)\right)$. To derive this equation, we use the fact that if an observable is given by the operator o(t) in the representation (H,h(t)) of the system S, then it is given by
(22)$$O\left(t\right):=\rho {\left(t\right)}^{-1}o\left(t\right)\rho \left(t\right),$$
in the representation $\left(H_{\eta (t)},H\left(t\right)\right)$, [12]. We also recall that the Heisenberg-picture operator corresponding to (Equation 22) has the form
(23)$$O^{\left(H\right)}\left(t\right):=U{\left(t,t_{0}\right)}^{-1}O\left(t\right)U\left(t,t_{0}\right).$$
In particular,
(24)$$H^{\left(H\right)}\left(t\right):=U{\left(t,t_{0}\right)}^{-1}H\left(t\right)U\left(t,t_{0}\right),$$
gives the expression for the Heisenberg-picture Hamiltonian in the representation $\left(H_{\eta (t)},H\left(t\right)\right)$. Furthermore, because $\rho \left(t\right):H_{\eta (t)}\to H$ maps the solutions of the Schrödinger for the Hamiltonian H(t) to those for h(t),
(25)$$U\left(t,t_{0}\right)=\rho {\left(t\right)}^{-1}u\left(t,t_{0}\right)\rho \left(t_{0}\right).$$
Equations (Equation 19) and (Equation 22)–(Equation 24) imply
(26)$$O^{\left(H\right)}\left(t\right)=\rho {\left(t_{0}\right)}^{-1}o^{\left(H\right)}\left(t\right)\rho \left(t_{0}\right).$$
Differentiating both sides of this equation with respect to *t* and making use of (Equation 21), (Equation 25), and the identity,
$$\rho {\left(t_{0}\right)}^{-1}h^{\left(H\right)}\left(t\right)\rho \left(t_{0}\right)-iU{\left(t,t_{0}\right)}^{-1}\rho {\left(t\right)}^{-1}\dot{\rho }\left(t\right)U\left(t,t_{0}\right)=H^{\left(H\right)}\left(t\right),$$
which follows from (Equation 16) and (Equation 25), we arrive at the Heisenberg equation in the representation $\left(H_{\eta (t)},H\left(t\right)\right)$:(27)$$i\partial_{t}O^{\left(H\right)}\left(t\right)=\left[O^{\left(H\right)}\left(t\right),H^{\left(H\right)}\left(t\right)\right]+iU{\left(t,t_{0}\right)}^{-1}\dot{O}\left(t\right)U\left(t,t_{0}\right).$$

Observe that because $\rho \left(t_{0}\right):H_{\eta (t_{0})}\to H$ is a unitary operator and $o^{\left(H\right)}\left(t\right):H\to H$ is Hermitian, (Equation 26) shows that $O^{\left(H\right)}\left(t\right)$ acts as a Hermitian operator in $H_{\eta (t_{0})}$. This is consistent with the basic requirement that for an evolving state vector ψ(t),
(28)$$\frac{{\langle \psi \left(t\right),O\left(t\right)\psi \left(t\right)\rangle }_{\eta (t)}}{{\langle \psi \left(t\right),\psi \left(t\right)\rangle }_{\eta (t)}}=\frac{{\langle \psi \left(t_{0}\right),O^{\left(H\right)}\left(t\right)\psi \left(t_{0}\right)\rangle }_{\eta (t_{0})}}{{\langle \psi \left(t_{0}\right),\psi \left(t_{0}\right)\rangle }_{\eta (t_{0})}}.$$

Comparing (Equation 21) and (Equation 27), we see that there is no structural difference between the Heisenberg equations for the representations (H,h(t)) and $\left(H_{\eta (t)},H\left(t\right)\right)$.

## 5. Identification of the Energy Operator

The conflict between the unitarity of dynamics and the observability of the Hamiltonian that appears in the representations of quantum system with a time-dependent Hilbert space shows that the Hamiltonian operator appearing in the standard Schrödinger equation does not coincide with the operator associated with the energy observable in these representations. The distinction between theses operators seems to disappear when the Hilbert space is static, simply because we are accustomed to follow the convention of identifying them. The above conflict provides a clear indication that this convention is not generally consistent. In the following, we argue that it is misleading even when the Hilbert space is time-independent.

Consider a quantum system S that is represented using a Hilbert space H with a constant inner product 〈·|·〉 and a Hermitian Hamiltonian operator *h* acting in H. The observables of S correspond to the Hermitian operator *o* acting in H. Now, consider a time-dependent unitary operator U(t) that maps H onto H. As is well-known, such an operator induces a quantum analog of a time-dependent classical canonical transformation. To see this, we recall that U(t) induces the following transformations on the state vectors Ψ∈H and the Hermitian operators o:H→H: (29)$$\begin{array}{cccc} \; & \Psi \to \tilde{\Psi }:=U\left(t\right)\Psi ,\; & \; & o\to \tilde{o}:=U\left(t\right)\;o\;U{\left(t\right)}^{-1}. \end{array}$$
These together with the fact that $U{\left(t\right)}^{†}=U{\left(t\right)}^{-1}$ ensure that the expectation values, 〈Ψ|oΨ〉/〈Ψ|Ψ〉, are invariant under these transformations. Therefore we can compute the kinematic properties of the system at any instant of time using either of Ψ and *o* or $\tilde{\Psi }$ and $\tilde{o}$. The same applies for the dynamical properties of the system provided that we postulate the following rule for the transformation of the Hamiltonian
(30)$$h\to \tilde{h}:=U\left(t\right)\;h\;U{\left(t\right)}^{-1}+i\;\dot{U}\left(t\right)\;U{\left(t\right)}^{-1}.$$
This ensures that Ψ(t) is a solution of the Schrödinger equation for the Hamiltonian *h* if and only if $\tilde{\Psi }\left(t\right):=U\left(t\right)\Psi \left(t\right)$ solves the Schrödinger equation for the Hamiltonian $\tilde{h}$.

Comparing (Equation 29) and (Equation 30), we see that under time-dependent quantum canonical transformations, the operators marking the observables of the system do not transform like the Hamiltonian operator (This is also true about the transformation property of the observables and the Hamiltonian in classical mechanics.) In particular, if we employ the convention of identifying the Hamiltonian *h* with the energy operator $h_{E}$, i.e., set $h_{E}=h$, we cannot do the same after we perform the time-dependent quantum canonical transformation induced by U(t); $h\to \tilde{h}$ while $h_{E}\to {\tilde{h}}_{E}=\tilde{h}-i\;\dot{U}\left(t\right)\;U{\left(t\right)}^{-1}\neq \tilde{h}$. This argument shows that we cannot consistently use this convention. In fact there seems to be no way of determining the energy operator, if we only know the Hamiltonian operator.

The additional structure that together with the Hamiltonian operator provide a consistent identification of the energy operator turns out to have a purely geometric nature [38]. The subtlety of dealing with time-dependent Hilbert spaces that we have examined in the preceding sections provides an important clue for uncovering this structure. The differential operator $D_{t}$ appearing on the left-hand side of the modified Schrödinger Equation (Equation 13) resembles a covariant time derivative with the term $\rho^{-1}\dot{\rho }$ reflecting the contribution of a local connection (gauge potential). According to (Equation 11) and (Equation 12) subtracting this term from the Hamiltonian operator gives the energy operator. Therefore, it seems that in order to identify a unique energy operator, we should look for an underlying vector (or principal) bundle E endowed with a connection [41,42,43]. Such a vector bundle has been constructed in Ref. [38] and used to formulate a geometric extension of quantum mechanics. The standard QM corresponds to situations where this bundle has a trivial topology. It is however important to recognize that topologically trivial vector bundles can possess nontrivial geometries. Indeed, it turns out that the determination of the energy observable is equivalent to the choice of a certain geometric structure, namely a metric-compatible connection, on this vector bundle.

## 6. Geometric Formulation of Quantum Dynamics

### 6.1. Vector Bundles

A vector bundle is a manifold E equipped with another manifold *M*, a function π mapping E onto *M*, and a vector space *V* such that the following conditions hold.

-There are open coordinate patches $O_{\alpha }$ covering *M* such that the subsets of E that are mapped into each of these patches, i.e.,
$$E_{\alpha }:=\left\{p\in E\;|\;\pi \left(p\right)\in O_{\alpha }\;\right\},$$
have the same topological structure as $O_{\alpha }\times V$. This means that for each $O_{\alpha }$, there is a continuous and invertible function $f_{\alpha }$ with a continuous inverse that maps $E_{\alpha }$ onto $O_{\alpha }\times V$.-For each R∈M, the points of E that are mapped to *R* by the function π form a vector space $V_{R}$.-For each R∈M and $p\in V_{R}$, let *v* be the element of *V* such that $f_{\alpha }\left(p\right)=\left(R,v\right)$. Then the function,
$$\phi_{\alpha ,R}:V_{R}\to V,$$
that is defined by $\phi_{\alpha ,R}\left(p\right):=v$ is a vector-space isomorphism, i.e., it is an invertible linear operator mapping $V_{R}$ onto *V*. In particular, $V_{R}$ and *V* are isomorphic vector spaces.

The manifolds E and *M* are called the total and base spaces, and the vector spaces *V* and $V_{R}$ are called the typical fiber and the fiber over *R*, respectively.

The basic motivation for the above definition of a vector bundle is actually very simple. Consider a pair of coordinate patches, $O_{\alpha }$ and $O_{\tilde{\alpha }}$, with a nonempty intersection. Then for each $R\in O_{\alpha }\cap O_{\tilde{\alpha }}$, we can use the so-called transition functions,
(31)$$g_{\tilde{\alpha }\alpha ,R}:=\phi_{\tilde{\alpha },R}\circ \phi_{\alpha ,R}^{-1},$$
to construct a one-to-one correspondence between the points of $O_{\alpha }\times V$ and $O_{\tilde{\alpha }}\times V$: (32)$$O_{\alpha }\times V∋\left(R,v\right)\overset{\;g_{\tilde{\alpha }\alpha ,R}\;}{⟷}\left(R,\tilde{v}\right)\in O_{\tilde{\alpha }}\times V\;\;\mathrm{if}\;\;\tilde{v}=g_{\tilde{\alpha }\alpha ,R}\left(v\right).$$
This correspondence allows us to reconstruct the total space of the vector bundle using the knowledge of the patches $O_{\alpha }$ of *M* and the transition functions $g_{\tilde{\alpha }\alpha ,R}$. To see this, we associate to each patch $O_{\alpha }$ and $R\in O_{\alpha }$ a vector space $V_{\alpha ,R}$ that is an identical copy of *V* and suppose that $V_{\alpha ,R}$’s with different (α,R) do not intersect, i.e., there is an isomorphism $\chi_{\alpha ,R}:V_{\alpha ,R}\to V$, and $V_{\alpha ,R}\cap V_{\alpha^{\prime },R^{\prime }}\neq ⌀$ if and only if $O_{\alpha }=O_{\alpha^{\prime }}$ and $R=R^{\prime }$. We also introduce
$$\begin{array}{ccc} V_{\alpha } & := & \underset{R\in O_{\alpha }}{⋃}V_{\alpha ,R},\\ E_{\alpha } & := & \left\{\left(R,v_{\alpha }\right)\in O_{\alpha }\times V_{\alpha }\;|\;v_{\alpha }\in V_{\alpha ,R}\;\right\}, \end{array}$$
and note that because $E_{\alpha }$ is an identical copy of $O_{\alpha }\times V$, we can use $\chi_{\alpha ,R}$ to identify $E_{\alpha }$ with $E_{\alpha }$. This observation together with the fact that $E={⋃}_{\alpha }E_{\alpha }$ suggests us to compare E with $E:={⋃}_{\alpha }E_{\alpha }$. These differ, because if $R\in O_{\alpha }\cap O_{\tilde{\alpha }}$ for some $\tilde{\alpha }\neq \alpha$, then to the fiber $V_{R}$ in E there corresponds two identical copies in E, namely $V_{\alpha ,R}$ and $V_{\tilde{\alpha },R}$. This shows that we can obtain E from E provided that we glue $V_{\alpha ,R}$ and $V_{\tilde{\alpha },R}$ along the intersections of the coordinate patches of *M*. Transition functions $g_{\tilde{\alpha }\alpha ,R}$ provide the missing gluing rule; we can use them to introduce the functions,
$${\overset{ˇ}{g}}_{\tilde{\alpha }\alpha ,R}:=\chi_{\tilde{\alpha },R}^{-1}\circ g_{\tilde{\alpha }\alpha ,R}\circ \chi_{\alpha ,R}:V_{\alpha ,R}\to V_{\tilde{\alpha },R},$$
and glue $V_{\alpha ,R}$ and $V_{\tilde{\alpha },R}$ according to the following prescription: $$V_{\alpha ,R}∋\left(R,v_{\alpha }\right)\;\mathrm{is}\;\mathrm{to}\;\mathrm{be}\;\mathrm{glued}\;\mathrm{to}\;\left(R,v_{\tilde{\alpha }}\right)\in V_{\tilde{\alpha },R}\;\;\mathrm{if}\;\;v_{\tilde{\alpha }}={\overset{ˇ}{g}}_{\tilde{\alpha }\alpha ,R}\left(v\right).$$

Because the transition functions are automorphisms of *V*, they belong to a subgroup *G* of the general linear group GL(V) of all automorphisms of *V*. The group *G* is called the structure group of the vector bundle.

If the fibers of E are complex (respectively real) vector spaces, E is called a complex (respectively real) vector bundle. If, as a manifold, E coincides with M×V, it is said to be a trivial vector bundle. For example $E_{\alpha }$ is a trivial vector bundle with base space $O_{\alpha }$, because it has the same topological structure as $O_{\alpha }\times V$. This shows that every vector bundle is locally trivial, for it can be expressed as the union of trivial vector bundles.

A smooth function ψ:M→E that maps every point *R* of *M* to a point in the fiber $V_{R}$ over *R* is called a global section of the bundle E. It turns out that if there are global sections $\psi_{1},\psi_{2},\cdots ,\psi_{N}$ such that for each R∈M, $\left\{\psi_{1}\left(R\right),\psi_{2}\left(R\right),\cdots ,\psi_{N}\left(R\right)\right\}$ is a basis of $V_{R}$, then E is a trivial bundle. The converse is also true if *V* is an *N*-dimensional vector space. For example, we can always construct such a collection of basis sections for the vector bundles $E_{\alpha }$. Because the domain of definition of these sections are not the whole base manifold but only one of its coordinate patches, namely $O_{\alpha }$, they are called local sections of E.

### 6.2. Parallel Transportation and Energy Operator

The geometry of a vector bundle E refers to a well-defined notion of parallel transportation of its points along curves in its base space *M*. This is achieved by an additional structure called a “connection.” We can reduce the problem of defining parallel transformation along curves in *M* to that for the segments of the curve that lie in particular patches of *M*. If we know how do define the parallel transportation of the points along each of these segments, we can pass from one patch to the adjacent one using the transition functions of the bundle. In the following we describe parallel transportation in a single patch.

Consider a coordinate patch $O_{\alpha }$ of *M*, and identify the points *R* of $O_{\alpha }$ with its real coordinates $\left(R^{1},R^{2},\cdots ,R^{d}\right)$. To characterize the points of the fibers we also introduce a fiber coordinate system. Suppose that *V* is a finite-dimensional complex vector space. Then without loss of generality we can identify it with $C^{N}$ for some $N\in Z^{+}$. Let $B:=\{e_{1},e_{2},\cdots ,e_{N}\}$ be the standard basis of $C^{N}$, i.e., $e_{m}:=\left(\delta_{m1},\delta_{m2},\cdots ,\delta_{mN}\right)$ where $\delta_{mn}$ is the Kronecker delta symbol. Because $\phi_{\alpha ,R}:V_{R}\to V=C^{N}$ is an isomorphism, $\phi_{\alpha ,R}^{-1}\left(e_{m}\right)$ form a basis of $V_{R}$. The functions $\psi_{m}:O_{\alpha }\to E_{\alpha }$ defined by
(33)$$\psi_{m}\left(R\right):=\phi_{\alpha ,R}^{-1}\left(e_{m}\right)$$
are examples of local sections of E that yield a basis of $V_{R}$ for each $R\in O_{\alpha }$, namely
$$B_{R}:=\left\{\psi_{1}\left(R\right),\psi_{2}\left(R\right),\cdots ,\psi_{N}\left(R\right)\right\}.$$
Given an element $v_{R}$ of $V_{R}$, we can expand it in this basis and use the coefficients of this expansion as the coordinates of $v_{R}$. In particular, if ψ:M→E is a global section of E, there are smooth functions $\Psi_{n}:O_{\alpha }\to C$ fulfilling
$$\psi \left(R\right)=\sum_{n=1}^{N}\Psi_{n}\left(R\right)\;\psi_{n}\left(R\right).$$

We may view the coefficient functions $\Psi_{n}$ as the components of a smooth vector-valued function $\Psi :O_{\alpha }\to C^{N}$ defined by
$$\Psi \left(R\right):=(\Psi_{1}\left(R\right),\Psi_{2}\left(R\right),\cdots ,\Psi_{N}\left(R\right)).$$
Let us now consider a basis transformation,
(34)$$\psi_{m}\left(R\right)\to \psi_{m}^{\prime }\left(R\right),$$
such that $\psi_{m}^{\prime }:O_{\alpha }\to E$ are also local sections whose values form a basis of $V_{R}$ for each $R\in O_{\alpha }$. If $\Psi_{m}^{\prime }:O_{\alpha }\to C$ are coefficients functions associated with the expansion of the global section ψ in the basis $B_{R}^{\prime }:=\left\{\psi_{1}^{\prime }\left(R\right),\psi_{2}^{\prime }\left(R\right),\cdots ,\psi_{N}^{\prime }\left(R\right)\right\}$, then (Equation 34) induces a linear coordinate transformation,
(35)$$\Psi_{m}\left(R\right)\to \Psi_{m}^{\prime }\left(R\right)=\sum_{n=1}^{N}gmn\left(R\right)\Psi_{n}\left(R\right),$$
where $g_{mn}:O_{\alpha }\to C$ are smooth functions whose values form the entries of an invertible matrix. Let $\Psi^{\prime }:O_{\alpha }\to C^{N}$ be the analog of Ψ that has $\Psi_{m}^{\prime }$ as its components. Then the coordinate transformation (Equation 35) is equivalent to
(36)$$\Psi \left(R\right)\to \Psi^{\prime }\left(R\right)=g(R)\left[\Psi \left(R\right)\right],$$
where $g:O_{\alpha }\to GL\left(n,C\right)$ is a smooth function, and $GL\left(N,C\right):=GL\left(C^{N}\right)$ is the general linear group of automorphisms of $C^{N}$. The functions Ψ provide local representations of the global sections ψ in $O_{\alpha }$. In the applications of vector bundles in particle physics, these describe the matter fields while the coordinate transformations (Equation 36) correspond to (local) gauge transformations.

Now, consider a smooth curve $\gamma :\left[t_{0},t_{1}\right]\to O_{\alpha }$ lying in $O_{\alpha }$, and identify γ(t) with its coordinates R(t). The parallel transportation of a point $\psi_{0}\in V_{R(t_{0})}$ along γ is a particular assignment of a point of $V_{R(t)}$ for each $t\in [t_{1},t_{2}]$. This defines a smooth curve $\Gamma_{A}:\left[t_{0},t_{1}\right]\to E_{\alpha }$. Because $\pi (\Gamma_{A}\left(t\right))=\gamma \left(t\right)$, $\Gamma_{A}$ is a lift of γ from $O_{\alpha }$ to $E_{\alpha }$. It is called the horizontal lift of γ. To determine it, we expand $\Gamma_{A}\left(t\right)$ in the basis $B_{R(t)}$, use $\Psi_{n}\left(t\right)$ to label the coefficients of this expansion, so that
$$\Gamma_{A}\left(t\right)=\sum_{n=1}^{N}\Psi_{n}\left(t\right)\psi_{n}\left[R\left(t\right)\right],$$
and identify $\Psi_{n}\left(t\right)$ with the solution of a homogeneous linear system of first-order differential equations. We can express this system in the form
(37)$$D_{t}\Psi \left(t\right)=0,$$
where
(38)$$D_{t}:=\partial_{t}+i\sum_{a=1}^{d}{\dot{R}}^{a}\left(t\right)A_{a}\left(R\left(t\right)\right),$$
and $A_{a}\left(R\right)$ are linear operators acting in $V=C^{n}$, i.e., they belong to the Lie algebra Gℓ(N,C) of the group GL(N,C). In physics literature, they are identified with the components of a gauge potential.

We can view $A_{a}\left(R\right)$ as the value of a smooth function $A_{a}:O_{\alpha }\to G\ell \left(n,R\right)$ and introduce a Gℓ(n,C)-valued one-form $A:=\sum_{a=1}^{d}A_{a}dR^{a}$ called a local connection one-form. Different choices of *A* determine different notions of parallel transformation in $E_{\alpha }$. Demanding that Equation (Equation 37) preserves its form under a gauge transformation (Equation 36), we are led to the following gauge transformation rule for local connection one-forms: $A\to A^{\prime }=\;gAg^{-1}-ig\;{dg}^{-1}$, where $dg:=\sum_{a=1}^{d}\partial_{a}gdR^{a}$ and $\partial_{a}$ stands for partial derivative with respect to $R^{a}$. Let us also note that the extension of the above procedure for parallel transformation to curves in *M* that do not lie in a single local coordinate patch requires patching together the horizontal lifts computed in adjacent patches, say $O_{\alpha }$ and $O_{\tilde{\alpha }}$, at an arbitrary point of the curve that lies in $O_{\alpha }\cap O_{\tilde{\alpha }}$. We can achieve this provided that at each $R\in O_{\alpha }\cap O_{\tilde{\alpha }}$ the local connection one-forms *A* and $\tilde{A}$, that are respectively associated with $O_{\alpha }$ and $O_{\tilde{\alpha }}$, are related via [41]
(39)$$\tilde{A}\left(R\right)=g_{\alpha \tilde{\alpha },R}^{-1}A\left(R\right)g_{\alpha \tilde{\alpha },R}-ig_{\alpha \tilde{\alpha },R}^{-1}\;dg_{\alpha \tilde{\alpha },R}.$$
If we can make a consistent assignment of local connection one-forms to all the patches $O_{\alpha }$ so that this equation holds in their intersection, we say that the vector bundle E is endowed with a connection A.

It is easy to see that we can express Equation (Equation 37) as the Schrödinger equation,
(40)$$i\partial_{t}\Psi \left(t\right)=H_{A}\left(t\right)\Psi \left(t\right),$$
for a Hamiltonian of the form
(41)$$H_{A}\left(t\right):=\sum_{a=1}^{d}A_{a}\left[R\left(t\right)\right]\;\partial_{t}R^{a}\left(t\right),$$
and identify its solution with
(42)$$\Psi \left(t\right)=U_{A}\left(t,t_{0}\right)\Psi \left(t_{0}\right),$$
where $U_{A}\left(t,t_{0}\right)$ is the evolution operator for $H_{A}\left(t\right)$.

An important property of the Hamiltonian (Equation 41) is that under smooth reparametrizations of *t*, i.e., $t\to t^{\prime }=\tau \left(t\right)$ for smooth monotonically increasing functions $\tau :[t_{0},t_{1}]\to R$, it transforms according to $H_{A}\left(t\right)\to H_{A}\left(t^{\prime }\right)={\left[\dot{\tau }\left(t\right)\right]}^{-1}H_{A}\left(t\right)$. This implies that such reparametrizations of time leave the Schrödinger Equation (Equation 40) and hence its solutions invariant. We can express the time-reparametrization invariance of solutions of (Equation 37) by expressing the time-ordered exponential yielding $U_{A}\left(t,t_{0}\right)$ as a path-ordered exponential along γ;
$$\begin{array}{ccc} U_{A}\left(t,t_{0}\right) & = & T\left\{\exp\int_{t_{0}}^{t}\;ds\left[-iH_{A}\left(s\right)\right]\right\}\\ & = & I+\sum_{\ell =1}^{\infty }{\left(-i\right)}^{\ell }\int_{t_{0}}^{t}ds_{\ell }\int_{t_{0}}^{s_{\ell }}ds_{\ell -1}\cdots \int_{t_{0}}^{s_{2}}ds_{1}H_{A}\left(s_{\ell }\right)H_{A}\left(s_{\ell -1}\right)\cdots H_{A}\left(s_{1}\right)\\ & = & I+\sum_{\ell =1}^{\infty }{\left(-i\right)}^{\ell }\int_{R(t_{0})}^{R(t)}A\left(R_{\ell }\right)\int_{R(t_{0})}^{R_{\ell }}A\left(R_{\ell -1}\right)\cdots \int_{R(t_{0})}^{R_{2}}A\left(R_{1}\right)\\ & = & P\left\{\exp\int_{R(t_{0})}^{R(t)}\;\left[-iA(R)\right]\right\}, \end{array}$$
where T and P respectively denote time-ordering and path-ordering operations, and the integrals over the Gℓ(n,C)-valued one-forms $A(R_{j})$ are to be performed along the segments of the curve γ.

The time-reparametrization invariance of the evolution operator for $H_{A}$ shows that the dynamics generated by this Hamiltonian in the typical fiber $C^{N}$ of the bundle $E_{\alpha }$ depends only on the shape of the curve γ and not on how fast this curve is traversed in time. In other words, it determines a purely geometrical evolution. Because this evolution yield a horizontal lift of γ, we call it a “horizontal evolution.”

We can also envisage more general lifts of γ that are associated with non-horizontal evolutions in the typical fiber. These would be determined by Hamiltonians $H\left(t\right):C^{N}\to C^{N}$ whose evolution operator does depend on the parameterization of the curve γ. The extreme situation is that of evolutions that take place in a single fiber of $E_{\alpha }$, i.e., when γ is a constant curve; $\gamma \left(t\right)=R_{0}$ for all $t\in [t_{0},t_{1}]$ and some $R_{0}\in O_{\alpha }$. In this case, the evolution of a point $\psi_{0}\in V_{R_{0}}$ maps it to
(43)$$\psi_{E}\left(t\right):=\sum_{n=1}^{N}\Psi_{n}\left(t\right)\;\psi_{n}\left(R_{0}\right),$$
where $\Psi_{n}\left(t\right)$ are components of the solution of the Schrödinger Equation (Equation 2) for a Hamiltonian $H_{E}\left(t\right):C^{N}\to C^{N}$. Because $\psi_{E}\left(t\right)\in V_{R_{0}}$, we call the time-evolution generated by $H_{E}\left(t\right)$ a “vertical evolution.”

The more general time-reparametrization non-invariant dynamics corresponds to an evolution generated by a Hamiltonian of the form,
(44)$$H\left(t\right)=H_{A}\left(t\right)+H_{E}\left(t\right).$$
In this case we can use (Equation 38) and (Equation 41) to express the Schrödinger equation,
(45)$$i\partial_{t}\Psi \left(t\right)=H\left(t\right)\Psi \left(t\right),$$
in the form
(46)$$D_{t}\Psi \left(t\right)=H_{E}\left(t\right)\Psi \left(t\right).$$

The modified Schrödinger Equation (Equation 13) that is proposed in Refs. [32,34] to circumvent the conflict between unitarity and the observability of time-dependent pseudo-Hermitian Hamiltonians is a special case of (Equation 46). If we consider the realistic situations where the time-dependence of the Hamiltonian and the energy operator is governed through their dependence on a set of real dynamical control parameters, which we can identify with coordinates *R* of points of a parameter space *M*, then η=η(R), ρ=ρ(R), and for R=R(t) we have $\dot{\rho }=\sum_{a=1}^{N}\partial_{a}\rho \;{\dot{R}}^{a}$. With the help of this relation, we can identify (Equation 14) with the special case of (Equation 38) that is given by the following choice for the local connection one-form.
(47)$$A=-i\rho^{-1}d\rho ,$$
where $d\rho :=\sum_{a=1}^{N}\partial_{a}\rho \;dR^{a}$. It is this choice that identifies the energy observable $H_{E}$ with the “Hamiltonian” for the modified Schrödinger Equation (Equation 13).

The above analysis suggests that we can keep using the term “Hamiltonian” for the generator of time evolutions *H* in the Schrödinger Equation (Equation 45), and identify the energy operator with the generator of vertical evolutions $H_{E}$. It is then clear that the knowledge of *H* is not sufficient to determine $H_{E}$ unless we also know $H_{A}$. Given that the latter is uniquely determined by the connection one-form *A*, we are led to a geometric formulation of quantum dynamics where we can identify the evolution of state vectors with certain trajectories in a trivial vector bundle $E_{\alpha }$ endowed with a local connection-one form *A*. Each such trajectory is a lift of a curve of control parameters of the system. It is determined by the choice of *A* and the energy operator $H_{E}$. We can relate the latter with an assignment of a linear operator $H\left(R\right):V_{R}\to V_{R}$ to each $R\in O_{\alpha }$, because we can specify $H_{E}$ in the form
(48)$$H_{E}\left(t\right)=\phi_{\alpha ,R(t)}\circ H\left(R\left(t\right)\right)\circ \phi_{\alpha ,R(t)}^{-1}.$$
We can view H as a function mapping $O_{\alpha }$ into another vector bundle which we describe after we elucidate the notion of “observable” in our vector bundle setting for QM.

We end this subsection by stressing that the choice (Equation 47) for *A* is not dictated by any basic physical principle. This choice follows from the requirement of identifying the Hamiltonian h(t) with the energy operator, but as we discussed above, this requirement violates the invariance of expectation values of the energy observable under time-dependent quantum canonical transformations.

### 6.3. Hermitian Vector Bundles, Unitarity, and Observables

If each of the fibers $V_{R}$ of a complex vector bundle E is equipped with an inner product ${\langle \cdot ,\cdot \rangle }_{R}$, we call E a Hermitian vector bundle. This inner product makes the fibers of E into an inner-product space. For cases where the fibers are finite-dimensional, they are Hilbert spaces parameterized by the points *R* of *M*. If the fibers $V_{R}$ are infinite-dimensional separable Hilbert spaces, E is called a Hilbert bundle. These turn out to be topologically trivial [44,45], but they may possess nontrivial geometries.

If a Hermitian vector bundle is endowed with a connection, parallel transportations of a pair of points belonging to a fiber may change their inner product. There are however a special class of connections on Hermitian bundles where this does not happen, i.e., parallel transportation along all curves preserves the inner product. Such a connection is called a metric-compatible or simply a metric connection.

Let us fix a coordinate patch $O_{\alpha }$ of *M* and use the local sections $\psi_{m}:O_{\alpha }\to E_{\alpha }$ defined by (Equation 33) together with the inner product on the fibers of $E_{\alpha }$ to construct an inner product on $V=C^{N}$ as follows.

First, we introduce
(49)$$\eta_{mn}\left(R\right):={\langle \psi_{m}\left(R\right),\psi_{n}\left(R\right)\rangle }_{R},$$
and identify $\eta \left(R\right):C^{N}\to C^{N}$ and ${\langle \cdot ,\cdot \rangle }_{\eta (R)}:C^{N}\times C^{N}\to C$ with the linear operator and inner product defined by
(50)$$\begin{array}{cccc} \; & \eta \left(R\right)w:=\sum_{m=1}^{N}\eta_{mn}\left(R\right)w_{n},\; & \; & {\langle \cdot ,\cdot \rangle }_{\eta (R)}:=\langle \cdot |\eta \left(R\right)\cdot \rangle , \end{array}$$
where $w:=(w_{1},w_{2},\cdots ,w_{N})$ is an arbitrary element of $C^{N}$, and 〈·|·〉 is the Euclidean inner product on $C^{N}$. Then, for every $v:=\left(v_{1},v_{2},\cdots ,v_{N}\right)\in C^{N}$, we have
(51)$$\begin{array}{ccc} {\langle v,w\rangle }_{\eta (R)} & = & \langle v|\eta \left(R\right)w\rangle =\sum_{m,n=1}^{N}v_{m}^{*}\eta_{mn}\left(R\right)w_{n}\\ & = & \sum_{m,n=1}^{N}v_{m}^{*}w_{n}{\langle \psi_{m}\left(R\right),\psi_{n}\left(R\right)\rangle }_{R}\\ & = & \sum_{m,n=1}^{N}v_{m}^{*}w_{n}{\langle \phi_{\alpha ,R}^{-1}\left(e_{m}\right),\phi_{\alpha ,R}^{-1}\left(e_{n}\right)\rangle }_{R}\\ & = & {\langle \phi_{\alpha ,R}^{-1}\left(v\right),\phi_{\alpha ,R}^{-1}\left(w\right)\rangle }_{R}. \end{array}$$
This calculation shows that ${\langle \cdot ,\cdot \rangle }_{\eta (R)}$ is a genuine inner product on $C^{N}$, and η is a metric operator acting in the Hilbert space $H:=(C^{N},\langle \cdot |\cdot \rangle )$. Furthermore, (Equation 51) implies that if we use $H_{\eta (R)}$ to denote the Hilbert space $\left(C^{N},{\langle \cdot ,\cdot \rangle }_{\eta (R)}\right)$, the isomorphisms $\phi_{\alpha ,R}:V_{R}\to H_{\eta (R)}$ are unitary operators. See Figure 1 for a schematic representation of the related mathematical constructs.

Next, suppose that E is provided with a connection A, and *A* is the corresponding local connection one-form on $E_{\alpha }$. Let $\gamma :\left[t_{0},t_{1}\right]\to O_{\alpha }$ be a smooth curve, R(t) label the coordinates of γ(t), and ϕ(t) and ψ(t) be elements of $V_{R(t)}$ that are respectively obtained by the parallel transportation of points $\phi_{0}$ and $\psi_{0}$ of $V_{R(t_{0})}$ along γ. By definition, A is a metric connection if for all choices of $O_{\alpha }$, γ, $\phi_{0}$, and $\psi_{0}$,
(52)$${\langle \phi \left(t\right),\psi \left(t\right)\rangle }_{R(t)}={\langle \phi_{0},\psi_{0}\rangle }_{R(t_{0})}.$$
If $\Phi_{n}\left(t\right)$ and $\Psi_{n}\left(t\right)$ are the coefficients of the expansion of ϕ(t) and ψ(t) in the local sections $\psi_{n}\left(R\left(t\right)\right)$, so that
$$\begin{array}{cccc} \; & \phi \left(t\right)=\sum_{n=1}^{N}\Phi_{n}\left(t\right)\psi_{n}\left(R\left(t\right)\right),\; & \; & \psi \left(t\right)=\sum_{n=1}^{N}\Psi_{n}\left(t\right)\psi_{n}\left(R\left(t\right)\right), \end{array}$$
and $\Phi :=(\Phi_{1},\Phi_{2},\cdots ,\Phi_{N})$ and $\Psi :=(\Psi_{1},\Psi_{2},\cdots ,\Psi_{N})$, we can use (Equation 49) and (Equation 50) to express (Equation 52) as
(53)$${\langle \Phi \left(t\right),\Psi \left(t\right)\rangle }_{\eta (t)}={\langle \Phi \left(t_{0}\right),\Psi \left(t_{0}\right)\rangle }_{\eta (t_{0})},$$
where η(t):=η(R(t)). Equations (Equation 42) and (Equation 53) show that the evolution operator $U_{A}\left(t,t_{0}\right)$ associated with the Hamiltonian $H_{A}\left(t\right)$ acts in the Hilbert space $H_{\eta (t)}$ as a unitary operator. That is horizontal evolutions defined by a metric connection in $H_{A}\left(t\right)$ are unitary. In particular, $H_{A}\left(t\right)$ satisfies (Equation 8). Equivalently,
(54)$$H_{A}{\left(t\right)}^{†}=\eta \left(t\right)H_{A}\left(t\right)\eta {\left(t\right)}^{-1}+i\dot{\eta }\left(t\right)\eta {\left(t\right)}^{-1}.$$

Now, consider a general lift of the curve γ that is determined by (Equation 44)–(Equation 46). Then the evolution operator $U(t,t_{0})$ defines a unitary operator acting in $H_{\eta (t)}$ if and only if the Hamiltonian H(t) satisfies (Equation 8). In view of (Equation 54), we can express this condition in the form
(55)$$H_{E}{\left(t\right)}^{†}=\eta \left(t\right)H_{E}\left(t\right)\eta {\left(t\right)}^{-1},$$
i.e., $H_{E}\left(t\right)$ acts as an η-pseudo-Hermitian operator in H and as a Hermitian operator in $H_{\eta (t)}$. As a result, its expectation values are real provided that we compute them using the inner product (Equation 50). This suggests that we can safely identify it with an observable of a unitary quantum system S that is represented by the pair $\left(H_{\eta (t)},H\left(t\right)\right)$ and call it the energy operator.

We can represent the quantum system S also using $\left(H,h(t)\right)$, where h(t) is given by (Equation 16). In view of this relation and (Equation 44), h(t) admits the decomposition: $$h\left(t\right)=h_{A}\left(t\right)+h_{E}\left(t\right),$$
where
(56)$$\begin{array}{cc} \; & h_{A}\left(t\right):=\rho \left(t\right)H_{A}\left(t\right)\rho {\left(t\right)}^{-1}+i\dot{\rho }\left(t\right)\rho {\left(t\right)}^{-1}, \end{array}$$
(57)$$\begin{array}{cc} \; & h_{E}\left(t\right):=\rho \left(t\right)H_{E}\left(t\right)\rho {\left(t\right)}^{-1}, \end{array}$$
and $\rho \left(t\right):=\sqrt{\eta (t)}$. It is not difficult to show that both $h_{A}\left(t\right)$ and $h_{E}\left(t\right)$ act as Hermitian operators in H. According to (Equation 57), $h_{E}\left(t\right)$ is the energy operator in this representation. Let us also note that the special choice (Equation 47) for the local connection one form *A* implies $H_{A}\left(t\right)=-i\rho {\left(t\right)}^{-1}\dot{\rho }\left(t\right)$. Substituting this equation in (Equation 56), we find $h_{A}\left(t\right)=0$. Therefore, it is only for this choice of *A* that h(t) coincides with the energy operator $h_{E}\left(t\right)$.

Next, we recall that the operator $\phi_{\alpha ,R}:V_{R}\to H_{\eta (R)}$ is unitary. Therefore, we can use it to construct another representation of the quantum system S where the state vectors at time *t* belong to the fiber $V_{R(t)}$, the observables measured at this time are given by Hermitian operators $O:V_{R(t)}\to V_{R(t)}$, and the dynamics corresponds to the lifts of the curve γ traced by the control parameters *R*. In particular, the evolving states ψ(t) are given by (Equation 43) with $\Psi_{n}$ being components of a solution Ψ of (Equation 46). It is not difficult to see that
(58)$$\psi \left(t\right)=\phi_{\alpha ,R(t)}^{-1}\left(\Psi \left(t\right)\right).$$
Solving this equation for Ψ(t) and substituting the result in (Equation 46), we can identify $\psi :\left[t_{0},t\right]\to E_{\alpha }$ with a solution of the evolution equation,
(59)$$iD_{t}\psi \left(t\right)=H\left(t\right)\psi \left(t\right),$$
where
(60)$$\begin{array}{cc} \; & D_{t}:=\phi_{\alpha ,R(t)}^{-1}\circ D_{t}\circ \phi_{\alpha ,R(t)} \end{array}$$
is called the covariant time-derivative corresponding to the metric connection on E, and
(61)$$\begin{array}{cc} \; & H\left(t\right):=\phi_{\alpha ,R(t)}^{-1}\circ H_{E}\left(t\right)\circ \phi_{\alpha ,R(t)} \end{array}$$
is a Hermitian operator acting in $V_{R(t)}$ that represents the energy observable of S.

The existence of a representation of S that uses the fibers of $E_{\alpha }$ as the Hilbert space of state vectors and identifies the Hermitian operators acting in these fibers with the observables suggests a natural extension where the possibly nontrivial Hermitian vector bundle E plays the role of its trivial subbundle $E_{\alpha }$. This leads to a proposal for a geometric extension of quantum mechanics that we examine in the next section.

## 7. Geometric Extension of Quantum Mechanics

Any attempt at extending QM must address both its kinematic and dynamical aspects (By kinematic aspects, we mean the definition of states, observables, and the meaning and implications of observing an observable when the system is in a given state. By dynamical aspects, we mean the prescription according to which the time-evolution of the states or observables of the system are determined.) In particular, it should clarify how it affects or alters the projection axiom. Obviously, the most conservative approach is to make sure this axiom holds in a more general setting. In trying to extend the description of a quantum system using a trivial Hermitian vector bundle to situations that the bundle has a nontrivial topology, this can be easily achieved, for a measurement of an observable takes place at a single instant of time. This observation together with the developments we have reported above lead to a natural geometric extension of quantum mechanics (GEQM) that we describe in the sequel.

The postulates of GEQM involve another vector bundle which we label by u(E). This is a real vector bundle with base space *M*. Its fiber $u_{R}$ over the point R∈M is the real vector space of Hermitian operators acting in the fiber $V_{R}$ of E. Its typical fiber is the vector space of Hermitian operators acting in $C^{N}$, which we can identify with the Lie algebra u(N) of the unitary group U(N) (Note that we can express the elements of U(N) in the form $e^{iX}$ where X is an N×N Hermitian matrix. Therefore, the elements of the Lie algebra u(N) are of the form iX. u(N) has the structure of a real vector space, because it is closed under matrix addition and scalar multiplication of matrices by real numbers. In physics literature, u(N) is identified with the real vector space of N×N Hermitian matrices, because as real vector spaces they are isomorphic.) The transition functions $g_{\alpha \tilde{\alpha },R}:u\left(N\right)\to u\left(N\right)$ of u(E) are given by the following relations [38].
(62)$$\begin{array}{cccc} \; & g_{\tilde{\alpha }\alpha ,R}\left(o\right):=G_{\tilde{\alpha }\alpha ,R}o\;G_{\tilde{\alpha }\alpha ,R}^{-1}, & \; & G_{\tilde{\alpha }\alpha ,R}:=\tilde{\rho }\left(R\right)\;g_{\tilde{\alpha }\alpha ,R}\;\rho {\left(R\right)}^{-1}, \end{array}$$
where α and $\tilde{\alpha }$ label pairs of intersecting coordinates charts, *R* belongs to their intersection, $\rho \left(R\right)=\sqrt{\eta (R)}$, $\tilde{\rho }\left(R\right)=\sqrt{\tilde{\eta }\left(R\right)}$, $\eta \left(R\right):C^{N}\to C^{N}$ is the metric operator associated with the coordinate chart $O_{\alpha }$, which we introduced in Section 6.3, $\tilde{\eta }\left(R\right)$ is its analog for the coordinate chart $O_{\tilde{\alpha }}$, and $g_{\alpha \tilde{\alpha },R}$ are the transition functions of E.

Having introduced u(E), we can present the postulates of GEQM as follows.

A quantum system S is determined by a complex Hermitian vector bundle E endowed with a metric connection A, a global section H:M→u(E) of the vector bundle u(E), and a smooth parameterized curve $\gamma :[t_{0},t_{1}]\to M$, where the parameter of γ is time, $\left[t_{0},t_{1}\right]$ is the time interval in which we wish to describe the system, and *M* is the base space of E whose points correspond to a collection of classical external control parameters.The (pure) states of S at a time *t* are given by one-dimensional subspaces (rays) of the fiber $V_{R(t)}$ of E, where R(t) labels the value of γ at *t*. These are uniquely determined by nonzero elements of $V_{R(t)}$ which we identify with the state vectors of S at time *t*.The observables of S are represented by global sections O:M→u(H) of u(E). For a measurement of O at time *t*, one implements von-Neumann’s projection axiom for the operator O(R(t)), which acts as a Hermitian operator in $V_{R(t)}$. In particular, if the system is in the state given by a state vector $\psi \in V_{R(t)}$, the measurement yields a reading that is an eigenvalue ω(t) of O(R(t)) and causes an abrupt change of the state of the system to one given by an eigenvector of O(R(t)) with eigenvalue ω(t). The probability of reading ω(t) and the expectation value of O(R(t)) are computed using the textbook prescription with $V_{R(t)}$ and O(R(t)) respectively playing the roles of the Hilbert space and the operator representing the observable.The evolution of the state vectors ψ(t) are determined by the covariant Schrödinger equation,
(63)$$iD_{t}\psi \left(t\right)=H\left(R\left(t\right)\right)\psi \left(t\right),$$
where $D_{t}$ is the covariant time-derivative defined by the connection, and H is the global section of u(E) that represents the energy observable.

It is not difficult to check that whenever the curve γ lies in a single coordinate patch of E, we can describe the system using $E_{\alpha }$. In this case we recover the representation of the system we outlined in Section 6.3. In particular, we can represent the system in terms of the Hilbert space H and the Hamiltonian h(t) using the standard rules of QM. This shows that GEQM reduces to QM locally. The same is the case if E happens to be a trivial bundle. In general, however, E is nontrivial, and we find an extension of QM. At present the physical implications of the structural differences between GEQM and QM are not clear.

It is a well-known mathematical fact that whenever the typical fiber of a vector bundle E is an infinite-dimensional Hilbert space, it is necessarily trivial [44,45]. This suggests that GEQM and QM are different only for systems with finite-dimensional state spaces (For a specific example of a class of toy models with two-dimensional state spaces see [38]).

The assertion that GEQM and QM coincide for situations where E is trivial may seem as a negative result, but we should realize that topologically trivial vector bundles can possess nontrivial geometries. This reveals a hidden geometric aspect of QM that is directly linked with the problem of identifying the energy operator.

## 8. Heisenberg Picture of Dynamics in GEQM

In the preceding section we have offered a description of GEQM in which the state vectors undergo dynamical evolutions. For a given observable represented by a global section O:M→u(H) of u(H), the expectation value of O for a measurement conducted at time *t* is given by
$$\frac{{\langle \psi \left(t\right),O\left(R\left(t\right)\right)\psi \left(t\right)\rangle }_{R(t)}}{{\langle \psi \left(t\right),\psi \left(t\right)\rangle }_{R(t)}},$$
where ${\langle \cdot ,\cdot \rangle }_{R(t)}$ is the inner product of the fiber $V_{R(t)}$ and $\psi \left(t\right)\in V_{R(t)}$ is the state vector at time *t*.

Suppose that the curve γ lies in a single coordinate patch $O_{\alpha }$ of *M*. Then we can use the unitary transformation $\phi_{\alpha ,R}:V_{\alpha }\to H_{\eta (R)}$ to introduce the operator,
(64)$$O\left(R\right):=\phi_{\alpha ,R}\;O\left(R\right)\phi_{\alpha ,R}^{-1},$$
which acts as a Hermitian operator in $H_{\eta (R)}$. Let us also recall that we determine ψ(t) from $\Psi \left(t\right):=\phi_{\alpha ,R(t)}\psi \left(t\right)$ and that Ψ(t) satisfies the Schrödinger equation defined by the Hamiltonian operator H(t) in the Hilbert space $H_{\eta (t)}$, where η(t):=η(R(t)).

Because $\phi_{\alpha ,R}:V_{\alpha }\to H_{\eta (R)}$ is unitary,
(65)$$\begin{array}{ccc} \frac{{\langle \psi \left(t\right),O\left(R\left(t\right)\right)\psi \left(t\right)\rangle }_{R(t)}}{{\langle \psi \left(t\right),\psi \left(t\right)\rangle }_{R(t)}} & = & \frac{{\langle \phi_{\alpha ,R(t)}\psi \left(t\right),\phi_{\alpha ,R(t)}\;O\left(R\left(t\right)\right)\psi \left(t\right)\rangle }_{\eta (R(t))}}{{\langle \phi_{\alpha ,R(t)}\psi \left(t\right),\phi_{\alpha ,R(t)}\psi \left(t\right)\rangle }_{\eta (R(t))}}\\ & = & \frac{{\langle \Psi \left(t\right),O\left(t\right)\Psi \left(t\right)\rangle }_{\eta (t)}}{{\langle \Psi \left(t\right),\Psi \left(t\right)\rangle }_{\eta (t)}}\\ & = & \frac{{\langle \Psi \left(t_{0}\right),O^{\left(H\right)}\left(t\right)\Psi \left(t_{0}\right)\rangle }_{\eta (t_{0})}}{{\langle \Psi \left(t_{0}\right),\Psi \left(t_{0}\right)\rangle }_{\eta (t_{0})}}\\ & = & \frac{{\langle \phi_{\alpha ,R(t_{0})}\psi \left(t_{0}\right),O^{\left(H\right)}\left(t\right)\phi_{\alpha ,R(t_{0})}\psi \left(t_{0}\right)\rangle }_{\eta (R\left(t_{0}\right))}}{{\langle \phi_{\alpha ,R(t_{0})}\psi \left(t_{0}\right),\phi_{\alpha ,R(t_{0})}\psi \left(t_{0}\right)\rangle }_{\eta (R\left(t_{0}\right))}}\\ & = & \frac{{\langle \psi \left(t_{0}\right),O^{\left(H\right)}\left(t\right)\psi \left(t_{0}\right)\rangle }_{R(t_{0})}}{{\langle \psi \left(t_{0}\right),\psi \left(t_{0}\right)\rangle }_{R(t_{0})}}, \end{array}$$
where we have used (Equation 28) and (Equation 64), set O(t):=O(R(t)), and introduced:(66)$$O^{\left(H\right)}\left(t\right):=\phi_{\alpha ,R(t_{0})}^{-1}O^{\left(H\right)}\left(t\right)\;\phi_{\alpha ,R(t_{0})}.$$
This is a Hermitian operator acting in $V_{R(t_{0})}$, i.e., it belongs to $u_{R(t_{0})}$. In view of (Equation 23), we can express it in the form,
(67)$$O^{\left(H\right)}\left(t\right)=U{\left(t,t_{0}\right)}^{-1}O\left(R\left(t\right)\right)\;U\left(t,t_{0}\right),$$
where $U\left(t,t_{0}\right):V_{R(t_{0})}\to V_{R(t)}$ is the linear operator defined by
(68)$$U\left(t,t_{0}\right):=\phi_{\alpha ,R(t)}^{-1}U\left(t,t_{0}\right)\phi_{\alpha ,R(t_{0})},$$
and $U(t,t_{0})$ is the evolution operator for the Hamiltonian H(t).

It is easy to see that $\psi \left(t\right)=U\left(t,t_{0}\right)\psi \left(t_{0}\right)$. This together with (Equation 66) and (Equation 67) suggest identifying $O^{\left(H\right)}\left(t\right)$ with the Heisenberg-picture operator associated with the observable represented by the global section O. According to (Equation 66), we can identify $O^{\left(H\right)}\left(t\right)$ with the solution of the Heisenberg Equation (Equation 27) that satisfies the initial condition,
(69)$$O^{\left(H\right)}\left(t_{0}\right):=O\left(t_{0}\right)=\phi_{\alpha ,R(t_{0})}O\left(R\left(t_{0}\right)\right)\phi_{\alpha ,R(t_{0})}^{-1}.$$

If the curve $\gamma :[t_{0},t_{1}]\to M$ of the parameters of the system does not lie in a single coordinate patch, we can dissect it into segments belonging to coordinate patches. We can then integrate (Equation 27) to determine $O^{\left(H\right)}\left(t\right)$ and $O^{\left(H\right)}\left(t\right)$ for each segment and connect the solutions using the appropriate transition functions. To see the details of this procedure, suppose that γ consists of segments $\gamma_{0}:\left[t_{0},{\tilde{t}}_{0}\right]\to O_{\alpha }$ and $\gamma_{1}:\left[{\tilde{t}}_{0},t_{1}\right]\to O_{\tilde{\alpha }}$ where $O_{\alpha }$ and $O_{\tilde{\alpha }}$ are coordinate patches of *M* with $R\left({\tilde{t}}_{0}\right)\in O_{\alpha }\cap O_{\tilde{\alpha }}$, i.e.,
$$\gamma \left(t\right)=\left\{\begin{array}{ccc} \gamma_{0}\left(t\right) & \mathrm{for} & t\in [t_{0},{\tilde{t}}_{0}],\\ \gamma_{1}\left(t\right) & \mathrm{for} & t\in ({\tilde{t}}_{0},t_{1}], \end{array}\right.$$
and $\gamma_{0}\left({\tilde{t}}_{0}\right)=\gamma_{1}\left({\tilde{t}}_{0}\right)$. Then an initial state vector $\psi \left(t_{0}\right)\in V_{R(t_{0})}$ evolves according to
(70)$$\psi \left(t\right)=\left\{\begin{array}{ccc} U\left(t,t_{0}\right)\psi \left(t_{0}\right) & \mathrm{for} & t\in [t_{0},{\tilde{t}}_{0}],\\ \tilde{U}\left(t,{\tilde{t}}_{0}\right)U\left({\tilde{t}}_{0},t_{0}\right)\psi \left(t_{0}\right) & \mathrm{for} & t\in ({\tilde{t}}_{0},t_{1}], \end{array}\right.$$
where $U(t,t_{0})$ and $\tilde{U}\left(t,{\tilde{t}}_{0}\right)$ are respectively given by (Equation 68) and
$$\tilde{U}\left(t,{\tilde{t}}_{0}\right):=\phi_{\tilde{\alpha },R\left(t\right)}^{-1}\tilde{U}\left(t,{\tilde{t}}_{0}\right)\phi_{\tilde{\alpha },R\left({\tilde{t}}_{0}\right)},$$
$\tilde{U}\left(t,{\tilde{t}}_{0}\right)$ is the evolution operator associated with the Hamiltonian,
$$\tilde{H}\left(t\right):={\tilde{H}}_{\tilde{A}}\left(t\right)+{\tilde{H}}_{E}\left(t\right)=\sum_{a=1}^{d}{\tilde{A}}_{a}\left[R\left(t\right)\right]{\dot{R}}^{a}\left(t\right)+\phi_{\tilde{\alpha },R\left(t\right)}H\left(R\left(t\right)\right)\phi_{\tilde{\alpha },R\left(t\right)}^{-1},$$
and the initial time ${\tilde{t}}_{0}$, and ${\tilde{A}}_{a}$ are component of the local connection one-form $\tilde{A}$ in the patch $O_{\tilde{\alpha }}$ which fulfills (Equation 39).

Equation (Equation 70) suggests that the Heisenberg-picture operator $O^{\left(H\right)}\left(t\right):V_{R(t_{0})}\to V_{R(t_{0})}$ is to be given by (Equation 67) for $t\in [t_{0},{\tilde{t}}_{0}]$, and by
(71)$$\begin{array}{ccc} O^{\left(H\right)}\left(t\right) & := & {\left[\tilde{U}\left(t,{\tilde{t}}_{0}\right)U\left({\tilde{t}}_{0},t_{0}\right)\right]}^{-1}O\left(R\left(t\right)\right)\;\tilde{U}\left(t,{\tilde{t}}_{0}\right)U\left({\tilde{t}}_{0},t_{0}\right), \end{array}$$
for $t\in ({\tilde{t}}_{0},t_{1}]$. Note also that
(72)$$\tilde{U}\left(t,{\tilde{t}}_{0}\right)U\left({\tilde{t}}_{0},t_{0}\right)=\phi_{\tilde{\alpha },R\left(t\right)}^{-1}\tilde{U}\left(t,{\tilde{t}}_{0}\right)\;g_{\tilde{\alpha }\alpha ,R\left({\tilde{t}}_{0}\right)}U\left({\tilde{t}}_{0},t_{0}\right)\phi_{\alpha ,R(t_{0})}.$$

It is clear that for $t\in [t_{0},{\tilde{t}}_{0}]$, the operator $O^{\left(H\right)}\left(t\right)$ given by (Equation 23) satisfies (Equation 27). To determine the analog of (Equation 27) for $t\in [{\tilde{t}}_{0},t_{1}]$, we let
$$O^{\left(H\right)}\left(t\right):=\phi_{\alpha ,R(t_{0})}O^{\left(H\right)}\left(t\right)\phi_{\alpha ,R(t_{0})}^{-1},$$
and use (Equation 71) and (Equation 72) to show that, for $t\in [{\tilde{t}}_{0},t_{1}]$,
(73)$$O^{\left(H\right)}\left(t\right)=U{\left({\tilde{t}}_{0},t_{0}\right)}^{-1}g_{\tilde{\alpha }\alpha ,R\left(t_{0}\right)}^{-1}{\tilde{O}}^{\left(H\right)}\left(t\right)g_{\tilde{\alpha }\alpha ,R\left(t_{0}\right)}U\left({\tilde{t}}_{0},t_{0}\right),$$
where ${\tilde{O}}^{\left(H\right)}\left(t\right):=\tilde{U}{\left(t,{\tilde{t}}_{0}\right)}^{-1}\tilde{O}\left(t\right)\tilde{U}\left(t,{\tilde{t}}_{0}\right)$, $\tilde{O}\left(t\right):=\tilde{O}\left(R\left(t\right)\right)$, and
(74)$$\begin{array}{cc} \; & \tilde{O}\left(R\right):=\phi_{\tilde{\alpha },R}\;O\left(R\right)\phi_{\tilde{\alpha },R}^{-1}. \end{array}$$

Pursuing a similar approach as the one leading to (Equation 27), we can show that ${\tilde{O}}^{\left(H\right)}\left(t\right)$ satisfies the Heisenberg equation,
(75)$$i\partial_{t}{\tilde{O}}^{\left(H\right)}\left(t\right)=\left[{\tilde{O}}^{\left(H\right)}\left(t\right),{\tilde{H}}^{\left(H\right)}\left(t\right)\right]+i\tilde{U}{\left(t,{\tilde{t}}_{0}\right)}^{-1}\dot{\tilde{O}}\left(t\right)\tilde{U}\left(t,{\tilde{t}}_{0}\right),$$
and the initial condition ${\tilde{O}}^{\left(H\right)}\left({\tilde{t}}_{0}\right):=\tilde{O}\left({\tilde{t}}_{0}\right)$. According to (Equation 73), this implies that $O^{\left(H\right)}\left(t\right)$ satisfies (Equation 75) and the initial condition: (76)$$\begin{array}{ccc} O^{\left(H\right)}\left({\tilde{t}}_{0}\right) & = & U{\left({\tilde{t}}_{0},t_{0}\right)}^{-1}g_{\tilde{\alpha }\alpha ,R\left(t_{0}\right)}^{-1}\tilde{O}\left({\tilde{t}}_{0}\right)g_{\tilde{\alpha }\alpha ,R\left(t_{0}\right)}U\left({\tilde{t}}_{0},t_{0}\right)\\ & = & U{\left({\tilde{t}}_{0},t_{0}\right)}^{-1}O\left({\tilde{t}}_{0}\right)U\left({\tilde{t}}_{0},t_{0}\right), \end{array}$$
where we have employed (Equation 31), (Equation 64), and (Equation 74). Notice that (Equation 76) is consistent with the fact that for $t\in [t_{0},{\tilde{t}}_{0}]$, $O^{\left(H\right)}\left(t\right)$ satisfies (Equation 23). This in turn shows that $O^{\left(H\right)}\left(t\right)$ traces a smooth curve in the Hilbert space $H_{\eta (R\left(t_{0}\right))}$.

The procedure we have outlined for the cases where γ consists of a pair of segments each contained in a coordinate patch trivially extends to situations where it consists of an arbitrary number of such segments.

## 9. Concluding Remarks

Pseudo-Hermitian operators were initially considered in an attempt to provide a mathematically more careful assessment of some of the claims made by proponents of the importance of PT-symmetry, [6,7]. This clarified a number of issues of basic importance such as the spectral consequences of antilinear symmetries [8], the idea of reviving the Hermiticity of certain non-Hermitian operators by modifying the inner product of the Hilbert space [6,10,11], and a consistent definition of observables for PT-symmetric systems [12,20]. These developments involved considering time-independent pseudo-Hermitian Hamiltonian operators and led to various applications of these operators [5].

The study of time-dependent pseudo-Hermitian Hamiltonian operators was initially motivated by certain basic problems of quantum cosmology [22,23]. An important outcome of this study is a curious conflict between the unitarity of dynamics generated by such Hamiltonians and their observability [31]. This conflict has a more general domain of validity, for it applies to every quantum system whose state space is time-dependent. A proper resolution of this conflict calls for a more careful examination of the notion of energy operator for such systems. In this article, we have provided a geometric setting for addressing this issue, described the geometric meaning of the energy operator as the generator of vertical evolutions in a Hermitian vector bundle. A by-product of this approach is a consistent geometric extension of quantum mechanics. We have offered a general description of this extension and outlined the Heisenberg-picture formulation of its dynamical aspects. 

## Figures and Tables

**Figure 1 entropy-22-00471-f001:**
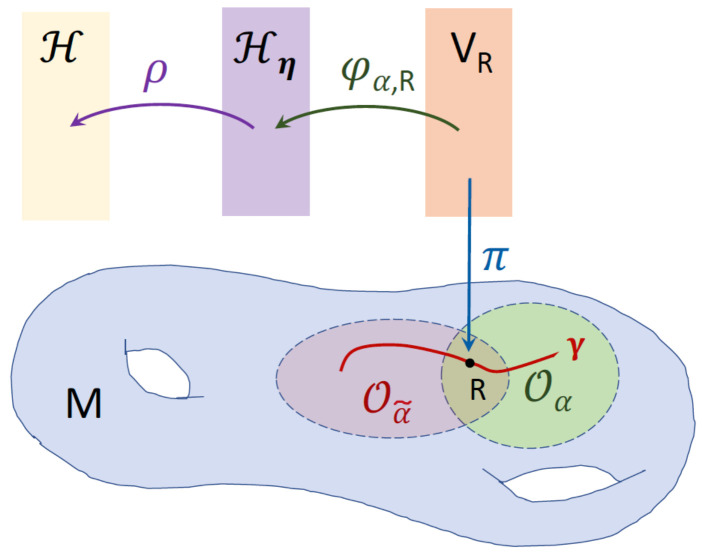
Schematic diagram representing the base space *M* of the vector bundle E, a curve γ in *M*, a pair of intersecting coordinate patches $O_{\alpha }$ and $O_{\tilde{\alpha }}$ of *M* that cover γ. *R* is a point in $O_{\alpha }\cap O_{\tilde{\alpha }}$. The function π:E→M is the bundle projection map that maps the fiber $V_{R}$ over *R* to *R*, i.e., $V_{R}=\pi^{-1}\left(\left\{R\right\}\right)$. $H_{\eta }$ and H are respectively the typical fiber $C^{N}$ endowed with the inner products ${\langle \cdot ,\cdot \rangle }_{\eta }$ and the Euclidean inner product 〈·|·〉. The isomorphisms $\varphi_{\alpha ,R}:V_{R}\to H_{\eta }$ and $\rho :H_{\eta }\to H$ are unitary operators.

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
