# Peer review of "Time-Dependent Pseudo-Hermitian Hamiltonians and a Hidden Geometric Aspect of Quantum Mechanics"

_entropy, 2020, doi:10.3390/e22040471_

Round 1

Reviewer 1 Report

In the manuscript "Time-Dependent Pseudo-Hermitian Hamiltonians and a Hidden Geometric Aspect of Quantum Mechanics" the author provides a detailed discussion of the fiber-bundle structures underlying systems described by time-dependent pseudo-Hermitian Hamiltonians what allows for a conceptually adequate understanding of these systems. Due to its novelty, conceptual depth and timeliness the material is clearly suitable for publication in "Entropy". Keeping in mind that "Entropy" has a broad readership of physicists who are not necessarily experts in the theory of non-Hermitian quantum mechanics and related operators, it would be good to slightly extend the manuscript by a number of clarifying comments for this broad non-expert readership.

Subsequently follows a list of corresponding suggestions.

1) page 2, lines 36,37: A clarifying comment could be added in the main text or in a further footnote making clear how an operator with real eigenvalues, but nontrivial Jordan block(s) in its matrix representation, is related to this basic result of linear algebra.

2) page 4, lines 95 - 102: To avoid confusion (for non-experts in the field) concerning the terminology of mappings being unitary it could make sense to add the concrete definition of "unitarity" as discussed, e.g., in the authors paper [A.M., Phys. Scr. 82:038110, 2010] and in [Reed, Simon, Functional Analysis, vol I,1980] and to make clear that this operator-theoretic definition (with its mapping between generally different Hilbert spaces) is broader than the definition of "unitarity" used in group theory (with group elements acting in the same Hilbert space representation).

3) page 12,13, lines 326-340: It appears not really plausible in line 330 why it should hold O_\alpha=O_{\alpha'} and R=R', and not R=R' \in O_\alpha \cap O_{\alpha'} as usually used in fiber bundle theory, i.e. with not necessarily coinciding patches O_\alpha and O_{\alpha'}. A reference to some textbook should be added where such a rather specific line of argumentation can be found.

4) page 17, line 467: To avoid readers' puzzling questions it could make sense to insert already here a comment on the differences between systems with finite dimensional Hilbert spaces and infinite dimensional ones.

5) page 20, line 522: For non-experts it could make sense to insert a clarifying comment what is meant by kinematic and dynamical aspects in the context of QM.

6) page 20, line 529-530: For non-experts it could make sense to slightly extend this description adding a comment making clear how elements of the Lie algebra u(N) are identified with elements from a real space (of Hermitian operators).

In addition the text contains a few minor typos:

7) page 5, line 116: "quantum mechanucs" ---> "quantum mechanics"

8) page 18, line 478: "and" ---> "an"

9) page 20, line 534: "introduce" ---> "introduced"

With the above suggestions implemented it can be expected that the manuscript will be suitable for immediate publication in "Entropy" (with its broad readership).

Reviewer 2 Report

please see attached

Reviewer 3 Report

I have carefully read the manuscript entropy-763284. The Author
provides a review of his work in the field of time-dependent
pseudo-hermitian hamiltonians. He reviews the problems arising
from the time evolution of a non-hermitian time-dependent
Hamiltonian. Particularly, he presents in a didactical and
rigorous form the fundamentals of a geometrical approach used
to resolve the conflict of unitarity and observability of a
time-dependent hamiltonian in the Heisenberg
representation (Geometric extension of Quantum Mechanics, GEQM).
In this framework, the Author provides a geometric meaning of
the energy operator and of the evolution operator. I find the manuscript gives an enlightening review about the
quantum dynamics of pseudo-hermitian hamiltonians for a general
audience. The contents are highly significant. I think that the
work has a relevant perspective, not only from the mathematical
theoretical background but also from the physical implications
in the study of different systems of interest. I have a couple of minor comments: 1- I think that, for the sake of completeness, some of the
works of A. Fring and collaborators should be included in the
bibliography, i.e. Physical Review A 93 (4), 042114, concerning
the proposal of finding unitary quantum evolution. 2- Concerning the by-product of construction of metric operators
for the time-independent case, some general proposal have been
given in Non-Selfadjoint Operators in Quantum Physics:
Mathematical Aspects edited by F. Bagarello, J. Gazeau,
F. Szafraniec, M. Znojil, Wiley-United States 2015, pag. 293 and
pag. 345, and in J. Math. Phys. 60, 012106 (2019). 3- In reading the manuscript I have found some typos: a) page 3 line 66.
b) page 5 line 117.
c) page 5 Eq.(7), third line.
d) Eq. after line 206 and Eq. (15).

From my previous observations, I recommend the manuscript for
publication in Entropy.  

Reviewer 4 Report

In this paper, the author reviews pseudo-Hermitian Hamiltonians with a time-dependent metric. For such Hamiltonians, there exists a contradiction, where the Hamiltonian cannot both generate unitary time evolution and be an observable thus necessarily violating one of the postulates of quantum mechanics. The author reviews several approaches to overcome this issue, such as modifying the Schrödinger equation, and finding metric operators that allow for unitary time evolution while the operators are not Hermitian, but particularly focuses on the arising of a geometric framework, which allows for a geometric extension of quantum mechanics.

The paper is well written and clearly presented, and presents an interesting overview on how to treat time-dependent metrics in the context of pseudo-Hermitian Hamiltonians. I have learned new things from reading the paper, and especially like the connection that is made to physics on occasion. As such, I recommend this paper for publication. However, as especially the second half of the paper is rather technical, I believe the manuscript would benefit from the inclusion of a figure or a table illustrating how the different notations are related to each other.

Additionally, I noticed several typos when reading the paper:

1. In line 40 it says “it generator the”, which should read “it generates the”.

2. In line 66, it says “viewed an an operator”, which should read “viewed as an operator”.

3. In line 71, it says “Hermitian operators acting”, which should read “Hermitian operators acting in”.

4. There is a small typo in line 117, the last word should read “mechanics”.

3. In line 151, it says “the followed spectral expansion”, which should read “the following spectral expansion”.

4. In the equation at the bottom of page 7, there are two psi’s on the left hand side, and a phi and a psi on the right hand side of the equality sign. This should be a phi and a psi on the left hand side as well.

5. In Eq.(18), the first equality reads U(t,t0) = U0(t,t0) exp(-i F(t)) should this not be exp(-i F(t) P)?

6. In line 277, it says “is time-dependent”. From the flow of the text, it seems that this should read “is time-independent”.

7. In the equation below Eq. (52), there is a psi on the right hand side of the equality sign for the first equation, which should be a phi.
